# Visible light-induced chemoselective 1,2-diheteroarylation of alkenes

Shi-Yu Guo[1], Yi-Peng Liu[1], Jin-Song Huang[1], Li-Bowen He[1,2], Gu-Cheng He[1,2], Ding-Wei Ji[1], Boshun Wan[1] & Qing-An Chen ●[1,2] ✉

Visible-light photocatalysis has evolved as a powerful technique to enable controllable radical reactions. Exploring unique photocatalytic mode for obtaining new chemoselectivity and product diversity is of great significance. Herein, we present a photo-induced chemoselective 1,2-diheteroarylation of unactivated alkenes utilizing halopyridines and quinolines. The ring-fused azaarenes serve as not only substrate, but also potential precursors for halogen-atom abstraction for pyridyl radical generation in this photocatalysis. As a complement to metal catalysis, this photo-induced radical process with mild and redox neutral conditions assembles two different heteroaryl groups into alkenes regioselectively and contribute to broad substrates scope. The obtained products containing aza-arene units permit various further diversifications, demonstrating the synthetic utility of this protocol. We anticipate that this protocol will trigger the further advancement of photo-induced alkyl/aryl halides activation.

The past decades have witnessed the evolution of visible-light photocatalysis[1–9] as a powerful tool to enable controllable radical reactions. Photocatalytic single electron transfer (SET)[10–17] and energy transfer (EnT)[18–24] are two principle modes of action to handle chemoselective reactions by generating active radical species (Fig. 1a). Among them, photoredox SET (oxidative or reductive quenching) is to induce an electron transfer to or from a reagent (A, D or S), thus generating the radical. The EnT process involves a photosensitizer transferring energy to a substrate (S), resulting highly reactive triplet state intermediate. Based on these instructions, significant strategies have been employed for divergent heteroarylation of alkenes under photocatalysis utilizing simple halopyridines or quinolines (Fig. 1b). For example, Jui's group developed anti-Markovnikov hydroarylation methods for alkenes by incorporating Hantzsch ester (HEH) as quencher via reductive quenching process[25–27]. In our previous work, an atom-economical halopyridylation of alkenes was achieved by introducing additional TFA to regulate the oxidative quenching pathway[28]. The protonated halopyridine as the switchable quencher could be directly reduced by exited state *Ir[III]. In addition, Glorius, Houk, Brown, etc. disclosed the EnT−mediated intermolecular dearomative cycloaddition reactions of quinolines with alkenes[29–31], where

quinolines activated by acid served as an effective quencher. Despite the significant progress made, developing unique photocatalytic modes for obtaining new chemoselectivity is of great significance, especially stemming from multicomponent halopyridine, alkene, and quinoline.

The rapid access to complex molecules from simple substrates is a crucial topic in synthetic chemistry. Polyarylalkanes are a class of important skeletons in natural products, pharmaceutical molecules, and advanced materials, which have attracted long-standing interests[32–39]. Recently, transition-metal catalyzed 1,2-diarylation of alkenes represents an effective strategy for the preparation of 1,2-diarylalkanes, representative works have been reported by Giri, Engle, Brown, Koh, Kong *etc*[40–48]. Li's group developed an electrochemical Co-catalyzed 1,2-diheteroarylation of styrenes enabled by dual C−H functionalizations of indoles[49]. However, substrates of the above works are usually restricted to alkenes that contain directing groups or conjugated aryls, and *N*-heteroaryl groups are also scarce because the coordinating nitrogen atom of azaaromatics tends to either deactivate the catalyst or interfere with the reaction selectivity. Hence, the catalytic 1,2-diheteroarylation of unactivated alkenes is expected to gain broad appeal but remains underdeveloped.

[1]Dalian Institute of Chemical Physics, Chinese Academy of Sciences, Dalian, China. [2]University of Chinese Academy of Sciences, Beijing, China. ✉e-mail: qachen@dicp.ac.cn

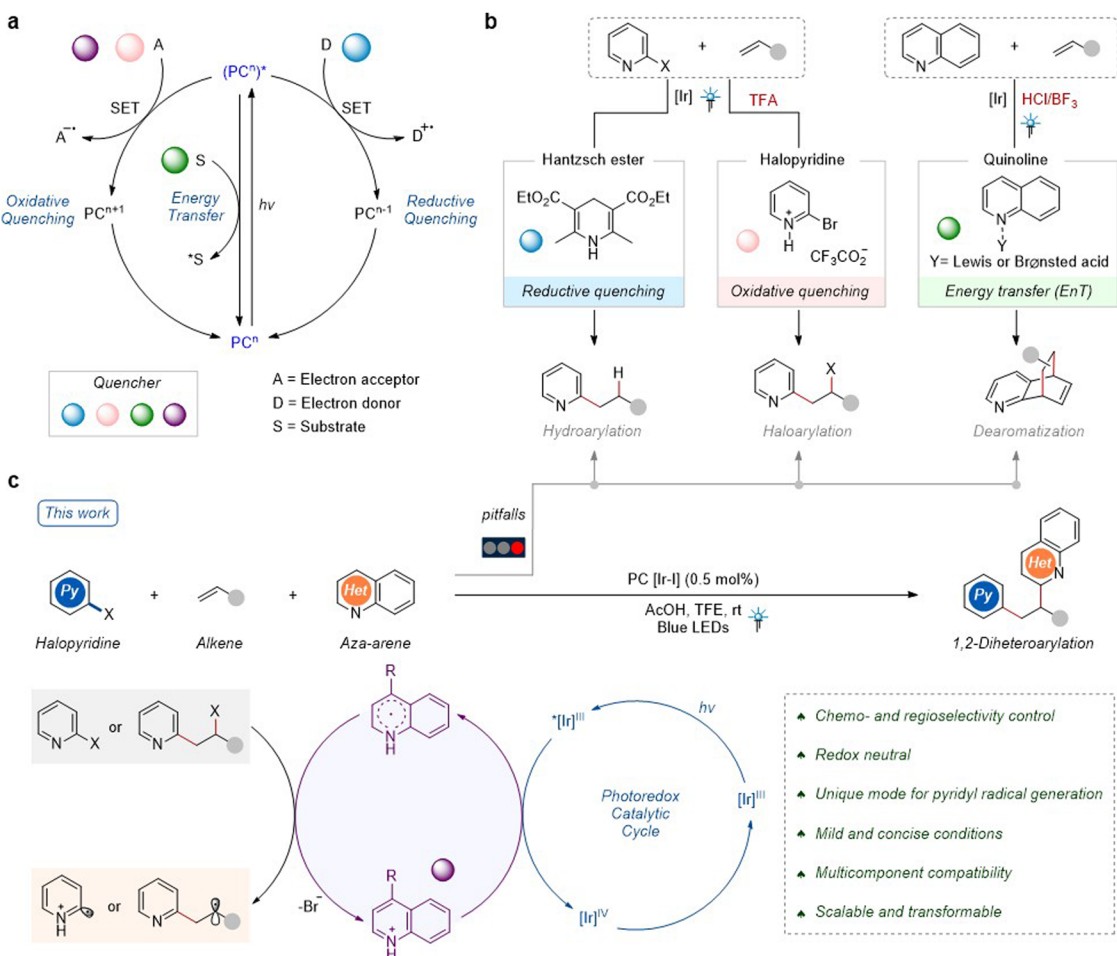

**Fig. 1 | Design plan for photocatalytic chemoselective 1,2-diheteroarylation of alkenes. a** Two principle modes of action in photocatalysis: SET and EnT. **b** Significant strategies for divergent heteroarylation of alkenes under photocatalysis. **c** This study: Visible light-induced chemoselective 1,2-diheteroarylation of alkenes. PC photocatalyst.

Given that complex products can be achieved by leveraging radical domino processes[50–53], photo-induced Minisci reactions[54–59] and our interest in photocatalytic (hetero)aryl halides activation[28,60,61], we questioned whether photocatalysis can address the challenges above and produce 1,2-diheteroarylalkanes from halopyridine, quinoline and alkene. In this context, it is challenging to keep the balance between the activation of halopyridines and ring-fused aza-arenes under the same acidic system, also not disturbing photocatalysis. The acidic system is a crucial factor but likely undergoes dynamic changes during the transformations. In addition, the effective quencher of excited-state photocatalysts is uncertain between halopyridine and ring-fused azaarene, which may affect the chemoselectivity. Herein, we present a visible light-induced chemoselective 1,2-diheteroarylation of alkenes from halopyridine, alkene, and ring-fused azaarene (Fig. 1c). Notably, the ring-fused azaarenes serve as not only substrate but also potential precursors for halogen-atom abstraction for the generation of pyridyl radicals.

## Results and discussion

Our investigation commenced with screening various Brønsted acid additives for the model reaction among 2-bromopyridine **1a**, TMS-substituted alkene **2a**, and 4-methylquinoline **3a**. To our delight, weak organic acids could lead to desired product **4aa** up to 90% yield in the condition of Ir(dtbbpy)(ppy)$_2$PF$_6$ (Ir-I) and TFE (2,2,2-Trifluoroethanol) under blue LEDs irradiation (Fig. 2a). But strong organic acids, such as TsOH and TFA gave bad performance, which is indispensable to our previous halopyridine activation. Exploration of certain photocatalysts

with varying properties was conducted, and product yields were correlated to redox properties of photocatalysts while unrelated to triplet state energy (Fig. 2b)[2]. These results implied a SET pathway rather than EnT process of this reaction. Moreover, ultraviolet-visible (UV-vis) absorption spectroscopy of all reactants and their mixtures demonstrated that photocatalyst Ir-I acted as the only absorbing species among the wavelength range covered by Kessil light source ($\lambda_{max}$ = 456 nm) (Fig. 2c). Subsequently, to understand the nature of this interaction, fluorescence quenching experiments and Stern-Volmer analysis were conducted (Fig. 2d). As an interesting result, the luminescence emission of photocatalyst was quenched effectively by [**3a** + AcOH] mixture instead of [**1a** + AcOH], indicating that an interaction between photocatalyst and [**3a** + AcOH] mixture might exist to promote this reaction. Meanwhile, the addition of strong acid TsOH could regulate 2-bromopyridine **1a** to be an effective quencher, but AcOH may not be acidic enough to activate 2-bromopyridine **1a** for interacting with the exited state [Ir-I]*. The results above hinted at an exclusive SET process between photocatalyst and activated quinoline.

Then, the impact of other parameters was also examined (Fig. 2e). The product yield of **4aa** decreased from 90% to 20% when AcOH was removed from the reaction condition (Fig. 2e, entries 1, 2). As expected, this reaction cannot proceed in the absence of either Iridium photocatalyst or blue light (Fig. 2e, entry 3). Interestingly, excess TsOH (2.0 eq.) as a replacement for AcOH changed the chemoselectivity, producing bromopyridylation side-product **5** in 41% yield with no product **4aa** observed (Fig. 2e, entry 4). Another strongly polar protic

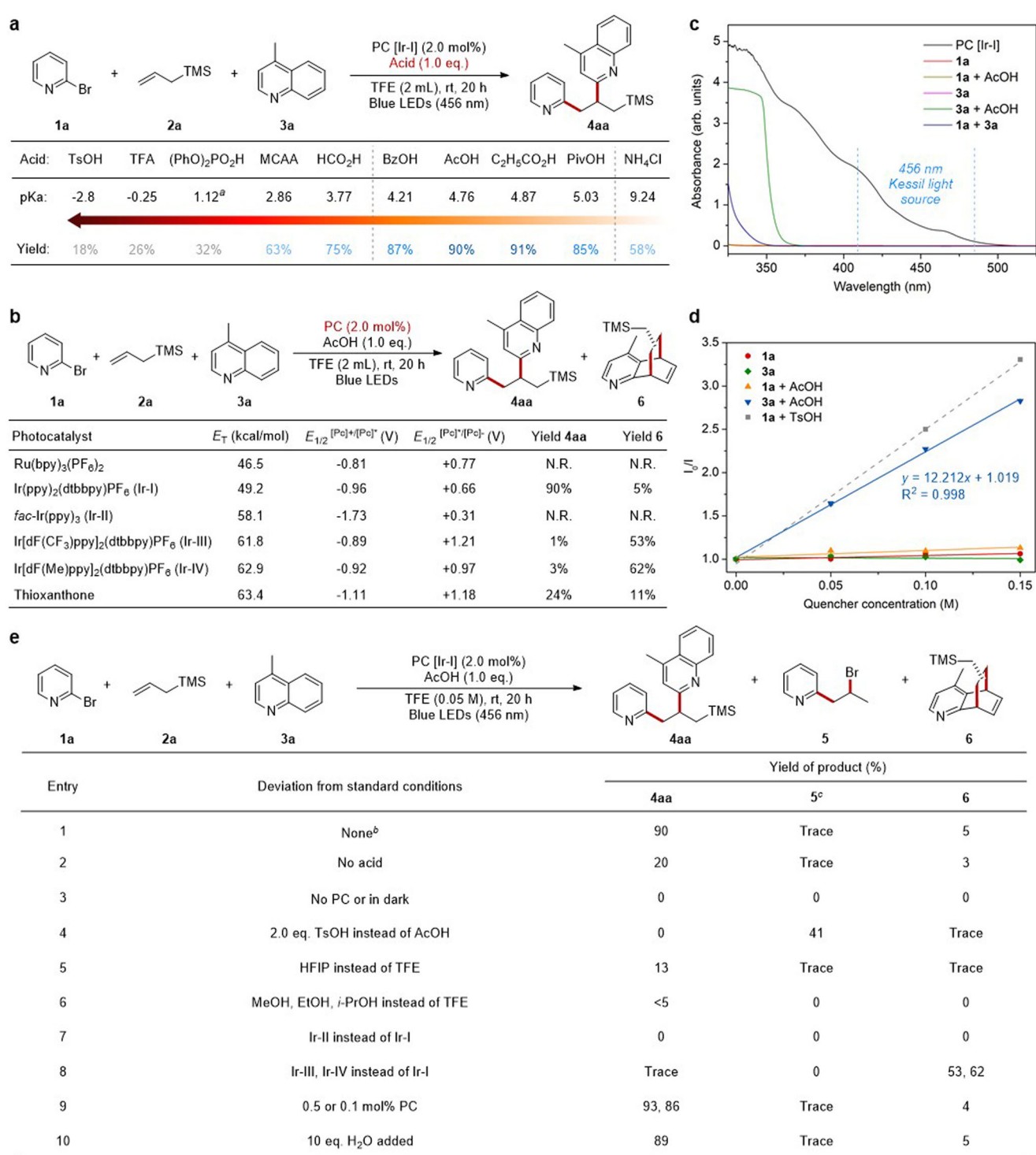

**Fig. 2 | Development of 1,2-diheteroarylation of alkenes via photoredox catalysis. a** Effect of Brønsted acid additives on model reaction. **b** Systematic evaluation of various photocatalysts. **c** UV-vis absorption spectroscopy. **d** Stern-Volmer quenching studies. **e** Evaluation of other reaction parameters. $^a$Predicted pKa value. MCAA chloroacetic acid. $^b$Reaction conditions: **1a** (0.20 mmol), **2a** (0.50 mmol), **3a** (0.10 mmol), PC [Ir-I] (2.0 mol%), AcOH (0.10 mmol), TFE (2.0 mL), blue LEDs ($\lambda_{max}$ = 456 nm), room temperature, $N_2$ atmosphere, 20 h, yields were determined by GC-FID analysis of the crude reaction mixture using 1,3,5-trimethoxybenzene as internal standard. $^c$Yields were calculated based on **1a**.

solvent HFIP (Hexafluoroisopropanol) afforded just 13% yield of product **4aa** (Fig. 2e, entry 5). Alcohols or common organic solvents, such as MeOH, EtOH, and MeCN, were all not feasible (Fig. 2e, entry 6 and Supplementary Table 3 in Supplementary information). Moreover, common organophotocatalysts, Ru-photocatalysts, and Ir(ppy)₃ (Ir-II) exhibited bad catalytic performance (Fig. 2e, entry 7 and Supplementary Table 1 in Supplementary information). Ir-photocatalysts with high triple-state energy, such as Ir[dF(CF₃)ppy]₂(dtbbpy)PF₆ (Ir-III) and Ir[dF(Me)ppy]₂(dtbbpy)PF₆ (Ir-IV) could switch the chemoselectivity to afford dearomatization side-product **6** with up to 62% yield (Fig. 2e, entry 8). The Ir(dtbbpy)(ppy)₂PF₆ (Ir-I) complex proved to be the most suitable catalyst for this reaction. To our satisfactory, product **4aa** could be obtained at 93% yield with the decreasing of photocatalyst dosage to 0.5 mol%, and still at 86% yield with only 0.1 mol%

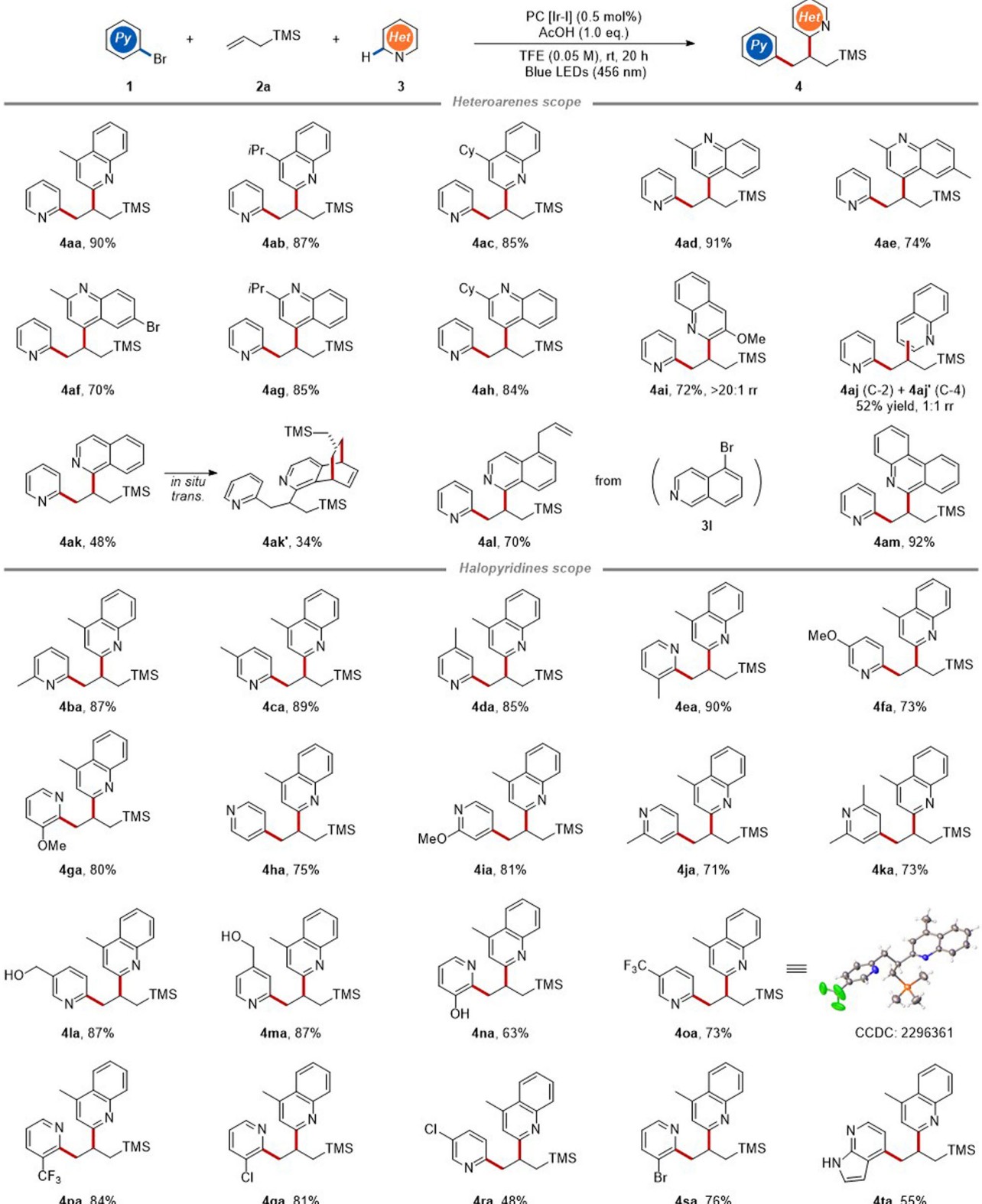

**Fig. 3 | Substrate scope of *N*-heteroaromatics.** Yields of isolated products. Regioselectivity (*rr*) was determined by ¹H NMR analysis.

photocatalyst (Fig. 2e, entry 9). This reaction appeared to be less sensitive to the water (Fig. 2e, entry 10).

With the optimized reaction conditions established, the generality of photocatalytic multi-component 1,2-diheteroarylation was explored, involving *N*-heteroaromatics scope (Fig. 3) and simple alkenes scope (Fig. 4). Initially, *N*-heteroarenes, including a series of quinolines (**4aa**–**4aj**), isoquinolines (**4ak** and **4al**) and phenanthridine (**4am**) were subjected to the optimized conditions (Fig. 3). The results demonstrated that quinolines bearing methyl or steric isopropyl, cyclohexyl substituents at either the C-2 or C-4 position proceeded

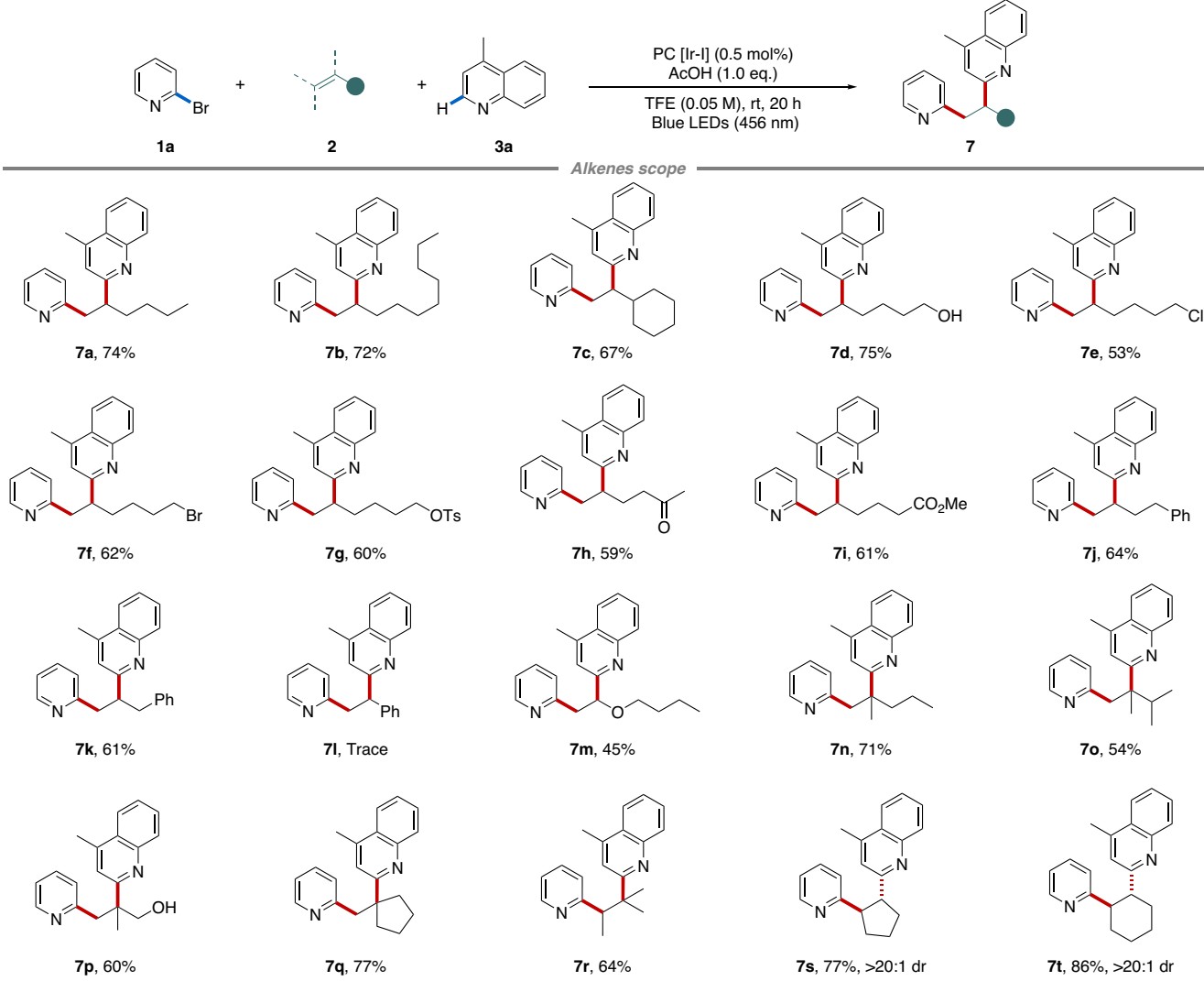

**Fig. 4 | Substrate scope of alkenes.** Yields of isolated products. Diastereoselectivity (*dr*) was determined by [1]H NMR analysis.

smoothly with good regioselectivities (**4aa**–**4ah**), while phenyl, ester and halo groups were not tolerated (Supplementary Fig. 1 in Supplementary information). In addition, 2-methyl-6-bromo-quinoline was an applicable heteroarene substrate, delivering corresponding product **4af** in 70% yield. The aforementioned results may be caused by electron-withdrawing substituents on *N*-containing aromatic rings altering the redox capacity of the quinoline, thereby disrupting the electron transfer. To our delight, the 3-methoxyquinoline exhibited good regioselectivity at C-2 position, affording the product **4ai** in 72% yield with >20:1 rr. Unsubstituted quinoline furnished mixtures of C-2/ C-4 regioisomers (**4aj** and **4aj'**) in comparable yields. Under optimized conditions, the isoquinoline-based product **4ak** (48% yield) was apt to overly react with alkene **2a**, leading to some further dearomative cycloaddition product **4ak'** (34% yield). Interestingly, when 5-bromoisoquinoline **3l** was subjected to this reaction, the inherent C −Br bond of resulting 1,2-diheteroarylation product was completely substituted by allyl group (**4al**). Reaction with phenanthridine could give product **4am** in a high yield (92%). Other aza-arenes such as pyridine, pyrazine, and pyrimidine were also explored, but afforded desired products in low yields (Supplementary Fig. 1 in Supplementary information).

Next, we turned to explore other halopyridines. As shown in Fig. 3, 2-bromopyridines substituted by methyl or methoxy at different positions performed well to produce target products **4ba**–**4ga** in

good to high yields under optimized conditions. A wide range of 4-bromopydines substrate reacted regioselectively at the C-4 position to deliver products in 71-81% yields (**4ha**–**4ka**). Bromopyridines containing sensitive functional groups, such as hydroxymethyl and hydroxyl, were also tolerated (**4la**–**4na**). Gratifyingly, **4oa** and **4pa** bearing trifluoromethyl as strong electron-withdrawing groups could also be isolated in 73% and 84% yields, respectively. The structure of **4oa** has been further confirmed by single-crystal X-ray crystallography (CCDC: 2296361). Moreover, dihalopyridines were also examined, and the reaction selectively took place at the C-2 position with the preservation of 3-Cl/Br (**4qa** and **4sa**) or 5-Cl (**4ra**) group for further synthetic elaboration. Notably, 4-Bromo-7-azaindole as diazo aryl halides was also reactive to afford 1,2-diheteroarylation product **4ta** in 55% yield.

The applicability of this protocol toward alkenes scope was investigated (Fig. 4). Simple α-olefins were successfully diheteroarylated with moderate to good yields (**7a**–**7k**). Among that, various functional groups, such as halide, benzenesulfonyl, acetyl, ester, and phenyl, were all feasible, but alkenes with electron-withdrawing substituents exhibited relatively low reactivity. Styrene was not suitable for this reaction (**7l**). Alternatively, enol ethers were also examined, but only delivered 45% yield of product **7m**. On the other hand, even 1,1-disubstituted alkenes (**7n**–**7q**) and internal alkenes (**7r**–**7t**) could be diheteroarylated regioselectively. For example, steric (**7n** and **7o**) or hydroxyl-substituted (**7p**) alkenes reacted smoothly to deliver

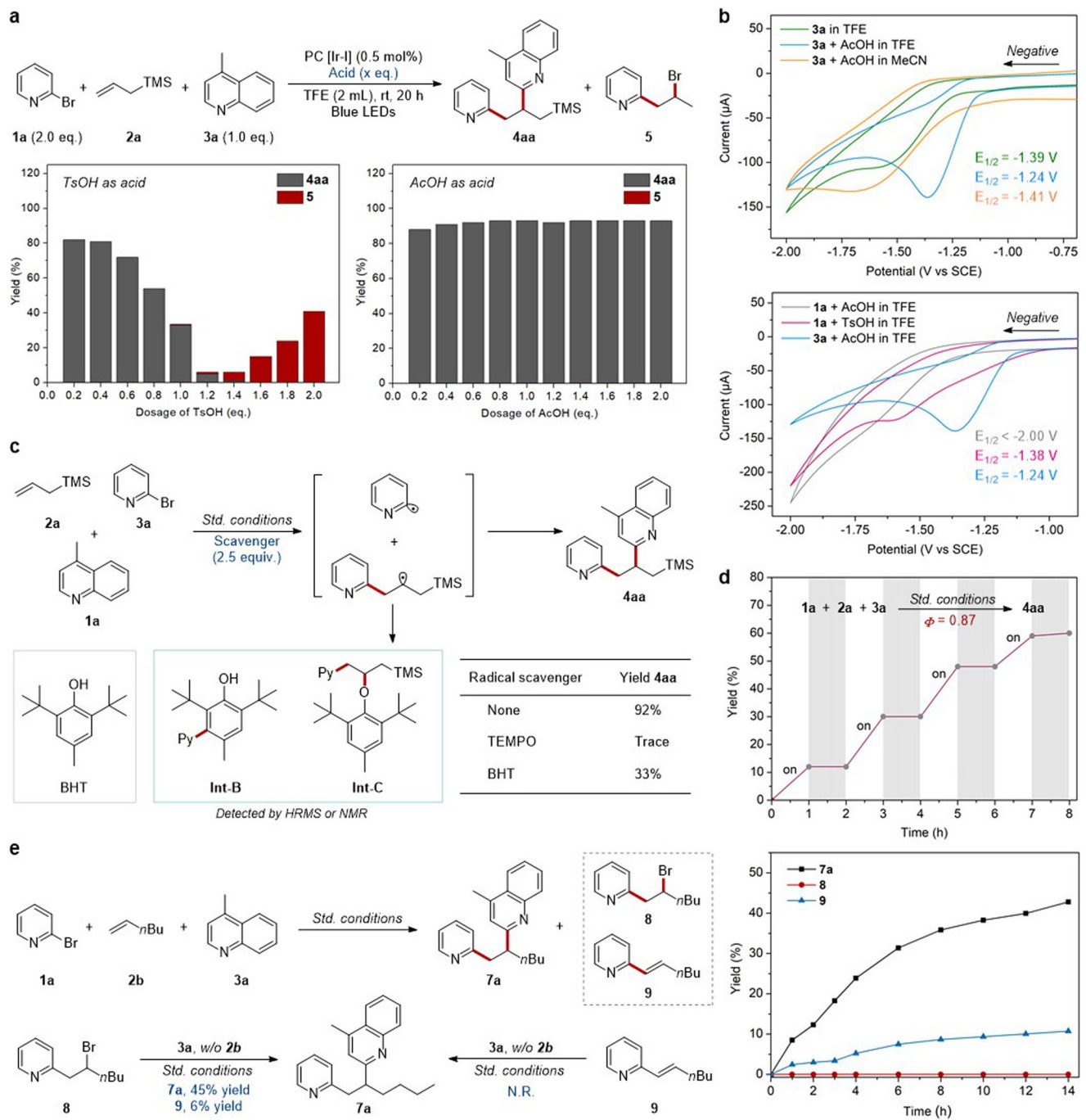

**Fig. 5 | Mechanistic investigations. a** Impact of varying amounts of additives on chemoselectivity. **b** Cyclic voltammetry studies. **c** Radical trapping experiments. **d** Light on/off experiments. **e** Kinetic studies and intermediates exploration. *Std.* standard.

corresponding products in 54–71% yields. Trisubstituted alkene participated in the production of a single product **7r** in 64% yield with high regioselectivity. Moreover, cyclopentene and cyclohexene as cyclic internal alkenes performed well, producing 77% (**7s**) and 86% (**7t**) yields of products, respectively, with excellent diastereoselectivities (>20:1 dr).

To gain deep insights into this chemoselective 1,2-diheteroarylation, more mechanistic studies were performed (Fig. 5). First, the impact of varying amounts (0.2 ~ 2.0 eq.) of TsOH or AcOH on the model reaction was examined respectively (Fig. 5a). With the increase of TsOH dosage, the chemoselectivity shifted from the desired product **4aa** to side-product **5**, while AcOH dosage has only a slight effect on yield of **4aa**, implying that the suitable strength or concentration of

Brønsted acids is of the utmost importance. These results are consistent with the Stern-Volmer quenching studies above (Fig. 2d). Second, the electrochemical analysis experiments were further conducted to understand the photoredox process. As shown in Fig. 5b, the cyclic voltammetry (CV) measurement of 4-methylquinoline **3a** indicated its reductive potential as −1.39 V vs. SCE in TFE (green line), and the addition of AcOH (1.0 equiv.) raised a sharper reduction peak of the [**3a** + AcOH] mixture to −1.24 V vs. SCE (blue line). Upon switching the solvent to MeCN, the reduction potential of the [**3a** + AcOH] mixture decreased to −1.41 V vs. SCE in MeCN (orange line). These results suggested that AcOH and TFE could effectively increase the redox potential of aza-arenes **3a** to be more easily reduced. Furthermore, no reduction peak was observed in the [**1a** + AcOH] mixture before −2.0 V

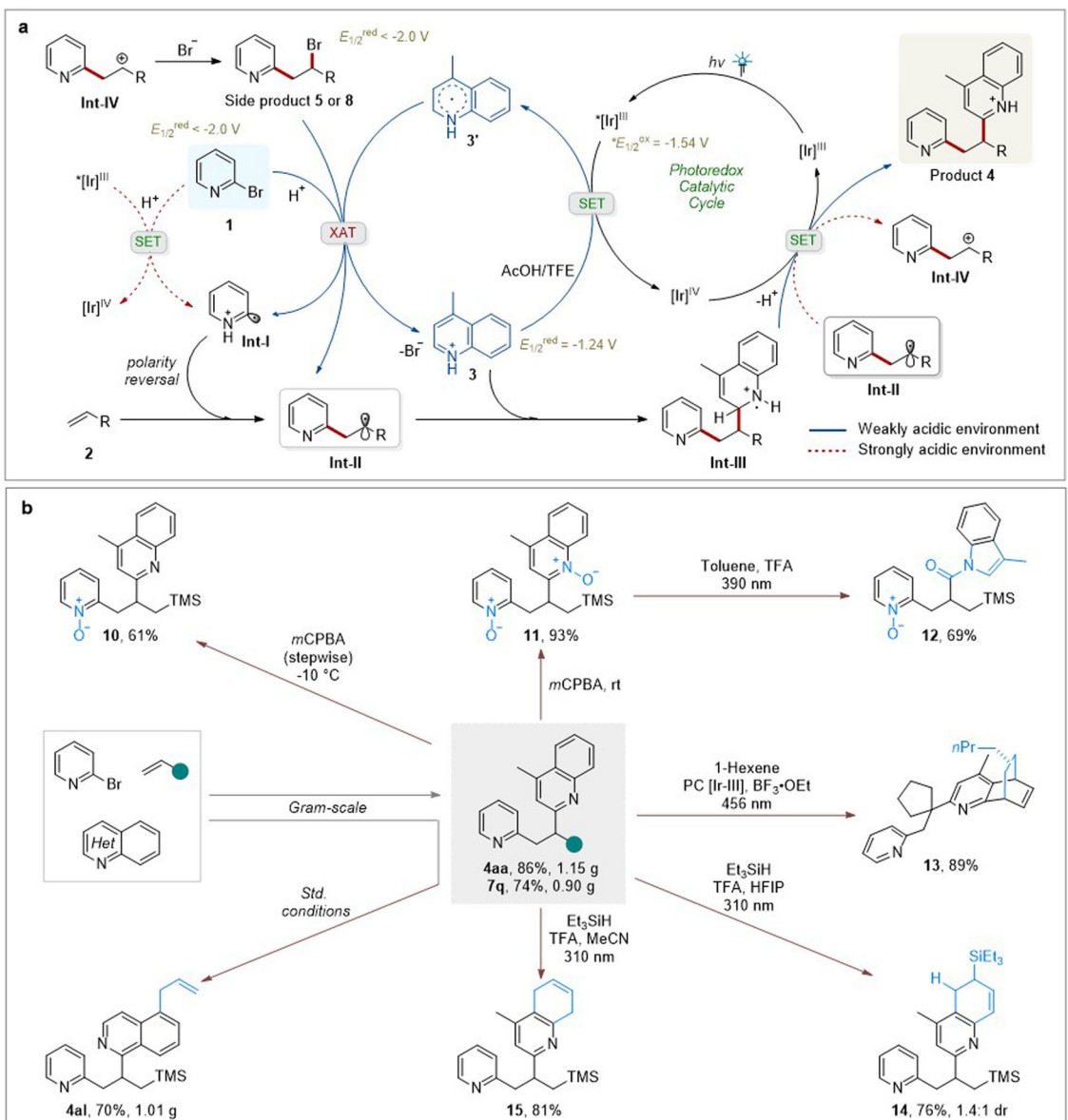

**Fig. 6 | Possible mechanism and synthetic transformations. a** Possible mechanism. **b** Synthetic transformations.

vs. SCE in TFE (gray line), while the mixture of [**1a** + TsOH] delivered a reduction potential as −1.38 V vs. SCE (pink line). Hence, it can be inferred that quinoline is more easily reduced than bromopyridine under AcOH/TFE conditions, while bromopyridine could also exhibit competitiveness if the acidity of the system is sufficiently strong.

Besides, the addition of radical scavenger TEMPO or BHT to the model reaction resulted in suppression of product generation (Fig. 5c). And two BHT-adducts, **Int-B** and **Int-C,** were successfully detected. These results indicate that pyridyl radical and secondary carbon-centered radical are probably formed in the transformation. Then, light on/off experiments showed that this transformation proceeded only under light irradiation, and the quantum yield of the model reaction was determined as 0.87, which might exclude the radical chain process (Fig. 5d). Notably, kinetic studies of **1a, 2b** and **3a** under standard conditions were tested (Fig. 5e). As more 1,2-diheteroarylation product **7a** generated, there was a slight accumulation of Heck-type side-product **9**, but with no observation of bromopyridylation side-product **8**. In the end, the pure **8** and **9** were resubjected to the standard conditions. As expected, **8** could react with **3a** to produce products **7a** and **9** in 45% and 6% yields, respectively, but **9** did not

react. So, it is inferred that bromopyridylation side-products may be generated in the transformation but will be quickly activated to produce target 1,2-diheteroaryl products. Besides, supplementary control experiments demonstrated that quinoline **3/3′** might interact with **1** to promote the generation of pyridyl radical (Supplementary Fig. 23 in Supplementary information).

Based on the combined results and related literatures[62–66], a possible mechanism is proposed as outlined in Fig. 6a. This reaction commences with the activation of photocatalyst Ir[III] to generate the excited state *Ir[III]. Then, under AcOH/TFE conditions, the SET between excited state *Ir[III] and protonated quinoline **3** creates oxidized Ir[IV] and heteroarene-centered radical **3′**. The excited state *Ir-I was reported as having $*E_{1/2}^{ox} = -0.96$ V (vs. SCE), which is a standard value in MeCN[67]. However, there will be a large variation in the redox property of photocatalysts in the presence of different solvents[68]. In this case, the experimental $*E_{1/2}^{ox}$ of Ir-I in TFE was measured as $*E_{1/2}^{ox} = -1.54$ V (vs. SCE) (Section 7.11 in Supplementary information). It suggests that the SET between *Ir[III] and protonated quinoline **3** ($E_{1/2}^{red} = -1.24$ V vs. SCE) is feasible. After that, the reduced radical **3′** is strongly nucleophilic like the α-aminoalkyl radical, which is able to promote the homolytic

activation of carbon-halogen bond[65,66]. In this protocol, intermediate **3'** might serve as a halogen abstracting reagent to activate bromopyridine **1**, producing the pyridyl radical **Int-I** (Supplementary Fig. 28 in Supplementary information). This assumption has been supported by the corresponding control experiments between 4-methylquinoline and alkyl halides (Supplementary Fig. 27 in Supplementary information). The electrophilic radical **Int-I** selectively couples with unactivated alkene **2** to furnish nucleophilic carbon-centered radical **Int-II**. Then, the obtained radical **Int-II** undergoes nucleophilic addition onto heteroarene **3** to generate radical cation **Int-III**. Finally, deprotonation and oxidation of **Int-III** by the Ir$^{IV}$ gives rise to target 1,2-diheteroaryl product **4** and regenerates photocatalyst Ir$^{III}$. In addition, an alternative SET between alkyl radical **Int-II** ($E_{1/2}^{Ox} = 0.47$ V)[69] and Ir$^{IV}$ (experimental $E_{1/2}^{red} = 0.98$ V vs. SCE, Supplementary Fig. 25 in Supplementary information) is electrochemically feasible to generate carbocation **Int-IV**, then affords bromopyridylation side-product **5** or **8**. Notably, based on the control experiment (Fig. 5e), alkyl halide **8** could also be activated to produce target 1,2-diheteroarylation product **7a**. The proposed mechanism for the generation of side-products **5**[28] and **6**[29] are also described (Section 7.10 in Supplementary information).

In order to further demonstrate the synthetic utility of this protocol, scale-up reactions alone with transformations of 1,2-diheteroaryl products were performed (Fig. 6b). Initially, 1,2-diheteroaryl products **4aa**, **7q**, and **4al** were successfully synthesized on 4.0 mmol scale without obvious loss of the yield. Then, the transformation was started with the oxidation of pyridine and quinoline moieties under *m*CPBA conditions[70]. Through stepwise addition of *m*CPBA at low temperature (−10 °C), pyridine single *N*-oxide **10** could be obtained in moderate yield. Bis-N-oxide **11** was also conveniently prepared in 93% yield, which underwent the subsequent photochemical carbon extrusion to deliver *N*-acylindole **12** via UV-light irradiation[71]. The intrinsic *N*-oxide reactivity could enable further diverse transformations of **10** and **12**. Moreover, photo-induced dearomatization of quinoline moiety of **4aa** and **7q** were accomplished, such as dearomative cycloaddition[29], hydrosilylation, and reduction[72], leading to partially saturated heteroarenes **13**–**15** in good yields (76–89%). These remaining double bonds after dearomatization represent another versatile, functional group to be further decorated.

In conclusion, we have demonstrated a visible light-induced catalytic 1,2-dihetearylation of unactivated alkenes in high chemo- and regioselectivities. The developed AcOH/TFE/Ir-I conditions are efficient for regulating the chemoselectivity through a unique mode for pyridyl radical generation, where the ring-fused aza-arenes serve as not only substrate but also precursors for potential halogen-atom abstraction in this photocatalysis. As a complement to metal catalysis methods, this photocatalytic radical process contributes to a broad substrate scope under mild conditions. In addition, the synthetic utility of the 1,2-diheteroaryl products was demonstrated in scale-up experiments and diverse transformations. A series of mechanistic studies were conducted to support the plausible mechanism. Further in-depth research and application are on the schedule in our group.

## Methods

### General procedures for chemoselective 1,2-diheteroarylation of alkenes under photocatalysis

To an oven-dried 20 mL vial was added Ir(ppy)$_2$(dtbbpy)PF$_6$ (0.002 mmol, 0.5 mol%), bromopyridine **1** (0.80 mmol, 2.0 eq.), alkene **2** (2.0 - 2.8 mmol, 5.0 ~ 7.0 eq.), quinoline **3** (0.40 mmol, 1.0 eq.), AcOH (0.40 mmol, 1.0 eq.) and TFE (8.0 mL, 0.05 M) in a nitrogen glove box. The vial was capped with a septum and wrapped with parafilm. The reaction mixture was stirred for 20 h under visible light irradiation (Kessil PR160, $\lambda_{max} = 456$ nm, 40 W, irradiation temperature maintained between 25–30 °C). Upon completion, the crude product was

neutralized with saturated NaHCO$_3$ solution or Et$_3$N and extracted with ethyl acetate. The organic layer was washed with brine solution and dried over anhydrous Na$_2$SO$_4$. Removal of the organic solvent in a vacuum rotavapor followed by flash silica gel column chromatographic purification afforded the desired product **4** or **7**.

## Data availability
Crystallographic data for the structures reported in this Article have been deposited at the Cambridge Crystallographic Data Center under deposition numbers CCDC 2296361 (**4oa**). Copies of the data can be obtained free of charge via https://www.ccdc.cam.ac.uk/structures/. Data relating to the characterization data of materials and products, general methods, optimization studies, experimental procedures, mechanistic studies, and NMR spectra are available in the Supplementary information. All data are also available from the corresponding author upon request.

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

## Acknowledgements
We acknowledge financial support from the National Natural Science Foundation of China (22301299 and 22271277), the Dalian Institute of Chemical Physics (DICP I202423), and the Doctoral Research Startup Fund Project of Liaoning Province (2020BS017).

## Author contributions
Q.-A. C. conceived and supervised the project. Q.-A. C., and S.-Y. G. designed the experiments. S.-Y. G., Y.-P. L., J.-S. H., L.-B. H., G.-C. H., D.-W. J., and B.-S. W. performed the experiments and analyzed the data. All authors discussed the results and commented on the manuscript.

## Competing interests
The authors declare no competing interests.
