## [Peer Review File · Nature Communications]

Visible Light-Induced Chemoselective 1,2-Diheteroarylation of AlkenesREVIEWER COMMENTS

Reviewer #1 (Remarks to the Author):

In the submitted manuscript, Chen and colleagues present an interesting approach involving a proton-coupled electron shuttle (PCES) for the chemoselective 1,2-diheteroarylation of inert alkenes under photocatalysis. This method employs halopyridines and quinolines. Notably, the PCES-mediated photocatalysis establishes a novel pathway for generating pyridyl radicals, thereby governing the chemoselectivity of the reaction. The resulting products exhibit intriguing heteroarene-linked structures, potentially contributing to the discovery of medicinally active compounds. Additionally, the authors discovered that an AcOH/TFE system, coupled with an appropriate photocatalyst, serves as a crucial switch, activating the quinoline-based PCES. In contrast, more acidic conditions deactivate it, leading to the formation of side products. These findings offer valuable mechanistic insights. As a complementary approach to metal catalysis, this photo-induced radical process operates under mild and redox-neutral conditions, allowing the selective incorporation of two distinct heteroaryl groups into alkenes and expanding the range of applicable substrates. The resulting products, containing aza-arene units, facilitate various subsequent modifications, highlighting the synthetic utility of this protocol. This methodology is poised to catalyze further advancements in the photo-induced activation of a broad scope of electrophiles and unsaturated substrates with controllable selectivity and will contribute to the development of other PCES-mode reactions. The reviewer supports the publication of this work in Nature Communications following the mentioned revisions below

1. The manuscript should elaborate on the intriguing generation of compounds 5 and 6 in the mechanism rationalization.
2. The authors are encouraged to properly cite relevant literature examples on Minisci reactivity using a photochemical approach, such as those found in ACS Catal. 2017, 7, 907; Chem. Eur. J. 2017, 23, 2537; and Science 2019, 363, 1429; Org. Lett. 2023, 25, 8541; Precis. Chem. 2023, 1, 437; Angew. Chem. Int. Ed. 2019, 58, 13666.
3. For the demonstrated azaarene for Minisci reactivity, the authors are suggested to demonstrate examples other than quinoline, such as substituted pyridine, pyrazine, and pyrimidine.

Reviewer #2 (Remarks to the Author):

Chen and co-workers report a strategy for the difunctionalization of olefins with two different heteroarenes. This transformation is potentially interesting with few similar reports in the literature. Although there are no direct applications shown of the method (e.g. to synthesis a known functional molecule), I found the synthetic aspect of the paper convincing enough to be at the correct level for Nat. Comms.

On the other hand, the mechanistic proposal and investigations leave a lot of questions to be answered:

- 1) First and foremost: why have the authors centred their manuscript around this so-called proton-coupled-electron-shuttle (PCES) concept? I cannot see either what this adds in terms of understanding, or indeed what this is supposed to be. At what point is the electron shuttle proton coupled? I see no analogies with PCET - in fact what seems to be suggested is just SET with a protonated heterocycle acting as a mediator (similar to electrochemistry). I

strongly suggest that the authors reconsider how they present this aspect.

2) Regarding the practicalities of this "PCES" - how does this work and how is it energetically favoured? In the proposed mechanism, if 3 is able to quench and undergo SET with Ir* to form 3' but 1 is not, how can 1 subsequently undergo SET with 3'? This does not make sense, unless there is some more nuance here regarding intramolecular interactions between the species (see mediators in electrochemistry).

3) In Fig. 5, the authors should add whether 8 and 9 can quench the excited state of the photocatalyst. This has implications for the mechanism that is suggested.

4) Certain parts of the text should be rephrased with an eye on precision. For instance:

(i) In the introduction, the authors assert with regards to SET and EnT that "in both modes, photocatalysts... directly interact with the substrate, following an "inner sphere-type" mechanism". This is not correct to the best of my knowledge.

(ii) "Notably, this PCES performs as a redox gear for photocatalyst." What does this mean?!

(iii) "Our investigations commenced with evaluating suitable acids that have compatible acidity coefficient" - what does this mean? That the pKa is above or below a certain value?

(iv) "...rules out a long radical chain process". I can infer what the authors are trying to say but more specific phrasing is necessary with regards to the on/off experiments, quantum yield and radical chain processes. As it stands, this generalisation does not make sense.

6) The authors should make clear what the photocatalysts, Ir-II, Ir-III etc. represent. The corresponding chemical formula should be given.

7) Based on the proposed mechanism, it is very hard to understand why styrene does not function in this reaction.

Ultimately, it feels like this manuscript is just an incremental advance over the previous publication from the group (10.1038/s41467-021-26857-w) with trapping of a radical generated by these products by a protonated heterocycle. If the mechanistic issues can be sufficiently resolved and the proposal of the authors convincingly supported then this manuscript could possibly be improved to the standard required for Nat. Comms. However, at the current time, I believe there are too many flaws to support publication.

Reviewer #3 (Remarks to the Author):

In this manuscript, Chen and coauthors disclose a visible-light photocatalytic chemoselective 1,2-diheteroarylation of unactivated alkenes with halopyridines and quinolines. In addition to the wide substrate range and versatility of the synthetic transformation, a series of experiments were performed to prove the proposed proton-coupled electron shuttle (PCES) for this new transformation. In my opinion, this is a nice paper worthy of publication in NatCommun after addressing the following issues:

1) A quantum yield should be provided because light on/off experiments are sometimes not sufficient to determine the reaction mechanism. Please refer to a publication by Yoon many years ago.

- 2) Please comment on the solvent effect and the unsuccessful substrate in S1.
 - 3) If possible, please isolate and confirm int-B and int-C, as well as the adduct of TEMPO with radical, because HRMS is not enough to confirm the proposed by-products in control experiments.
 - 4) Some NMR spectra do not seem to be pure enough or the sampling time is not long enough, such as 4ak, 4na, 4ra, 7i, etc.
- Anyway, this is a good work and it will deserve its publication in this excellent journal because of the good chemistry.

Response to Reviewer 1:

This reviewer recommended publication of this work in *Nature Communications* after some revisions. We gratefully appreciate the reviewer's appreciation on our work and respond to below.

Comments (1): The manuscript should elaborate on the intriguing generation of compounds 5 and 6 in the mechanism rationalization.

Response: Thanks very much for your suggestions. The producing pathway of side-products **5** and **6** was discussed in manuscript (Page 11, lines 3-4) and was shown in Supplementary information (Page S41, Supplementary Figure 22). Based on the mechanistic studies of our previous work (*Nat. Commun.* **2021**, *12*, 6538), the side-products **5** or **8** is probably generated through an acid-promoted domino photocatalytic oxidative quenching activation of halopyridines and radical-polar crossover pathway. When high triple-state energy photocatalyst is employed, photo-induced EnT for intermolecular dearomative cycloaddition of quinolines with alkenes occurs to produce side-product **6** (*Science* **2021**, *371*, 1338).

Comments (2): The authors are encouraged to properly cite relevant literature examples on Minisci reactivity using a photochemical approach, such as those found in ACS Catal. 2017, 7, 907; Chem. Eur. J. 2017, 23, 2537; and Science 2019, 363, 1429; Org. Lett. 2023, 25, 8541; Precis. Chem. 2023, 1, 437; Angew. Chem. Int. Ed. 2019, 58, 13666.

Response: Thanks very much for your suggestions. Several literatures about photo-induced Minisci reactions have been cited in our revised manuscript (Page 3, lines 4-5, refs. 54-59).

Comments (3): For the demonstrated aza-arene for Minisci reactivity, the authors are suggested to demonstrated examples other than quinoline, such as substituted pyridine, pyrazine, and pyrimidine.

Response: Thanks very much for your suggestions. Other aza-arenes, such as pyridine, pyrazine, pyrimidine *etc.* were also explored. As shown in Supplementary information (Page S7, Supplementary Figure 1c), although these substrates were reactive, but it afforded desired products in low yields compared with quinolines. And the corresponding comments were added to manuscript (Page 7, paragraph 1, lines 16-17).

Response to Reviewer 2:

This reviewer commented that the synthetic aspect of the paper is convincing enough to be at the correct level for *Nature Communications*, but further mechanistic investigations are required to explain the mechanism clearly. We gratitude the reviewer's appreciation on our work and we respond to below.

Comments (1): First and foremost: why have the authors centred their manuscript around this so-called proton-coupled-electron-shuttle (PCES) concept? I cannot see either what this adds in terms of understanding, or indeed what this is supposed to be. At what point is the electron shuttle proton coupled? I see no analogies with PCET - in fact what seems to be suggested is just SET with a protonated heterocycle acting as a mediator (similar to electrochemistry). I strongly suggest that the authors reconsider how they present this aspect.

Response: Thanks very much for your comments. In order to better present our work clearly, we revised it in the whole article (Title, abstract, introduction and conclusion *etc.*), for example:

a) The article title was changed to be “Visible Light-Induced Chemoselective 1,2-Diheteroarylation of Alkenes”;

b) In the abstract, we described the novelty of this methodology as the following sentences “We present a relay electron transfer strategy for chemoselective 1,2-diheteroarylation of unactivated

alkenes under photocatalysis utilizing halopyridines and quinolines. The ring-fused azaarenes serve as not only substrate but also an electron-transfer agent for pyridyl radical generation in this photocatalysis.”

Comments (2): Regarding the practicalities of this "PCES" - how does this work and how is it energetically favored? In the proposed mechanism, if 3 is able to quench and undergo SET with Ir to form 3' but 1 is not, how can 1 subsequently undergo SET with 3'?*

Response: Thanks very much for your comments. In order to better present our work clearly, we revised the term “PCES” as “Relay Electron Transfer”. We carried out supplementary control experiments to explain the mechanism. Although there is no direct evidence for **1** undergoing SET with **3'**, we think mechanistic investigations and related literatures are reasonable to support the proposed mechanism. Several experimental results support that quinoline **3** could be reduced by photocatalyst to promote pyridyl radical generation.

a) Fluorescence quenching experiments: The luminescence emission of photocatalyst was quenched effectively by [**3a**+AcOH] mixture instead of [**1a**+AcOH] (See manuscript, page 5, Figure 2d).

b) Cyclic voltammetry experiments: Quinoline **3a** (-1.24 V vs SCE) is more easily reduced than bromopyridine **1a** under AcOH/TFE conditions (See manuscript, page 9, Figure 5b).

c) Supplementary control experiments: Several control experiments were carried out to demonstrate the important role of quinoline **3/3'** for pyridyl radical generation. As shown in Supplementary information (Page S40), under AcOH/TFE standard conditions, **1a** and **2a** didn't react smoothly to produce halopyridylation product **5** or its elimination product **5'**. Replacing AcOH with stronger acid TFA gave 68% yield of **5** and 6% yield of **5'**. This result suggests that AcOH/TFE conditions are not acidic enough to activate 2-bromopyridine **1a** for interacting with PC. However, when 2,4-dimethylquinoline (1.0 equiv.) was added to AcOH/TFE standard conditions, it produced **5** and **5'** in 45% and 15% yields, respectively. A similar phenomenon was observed in the three multicomponent reaction. Substrate 2,4-dimethylpyridine **3n** just delivered 15% yield of 1,2-dihydroarylation product **4an** under AcOH/TFE conditions, while the addition of 2,4-dimethylquinoline (1.0 equiv.) could double the yield of **4an** (33%) accompanied with 17% yield of **5** and **5'**. Therefore, these above control experiments demonstrated that quinoline **3/3'** could interact with **1** to promote the generation of pyridyl radical. This conclusion was summarized in manuscript (Page 10, lines 21-22). Corresponding experimental studies and comments were presented in the supplementary information (Page S40, supplementary Figure 21).

There are some precedents about aza-arenes being reduced by reductant or photocatalyst, protonated aza-arenes being defined as electron-transfer agent in photoredox catalysis, and photocatalytic halogen-atom transfer. And these references were cited in the manuscript (Page 10, paragraph 2, line 1, ref. 62-65).

a) Jeffrey, J. L.; Petronijevic, F. R.; MacMillan, D. W. C. Selective radical-radical cross-couplings: Design of a formal β -mannich reaction. *J. Am. Chem. Soc.* **2015**, *137*, 8404;

b) Garza-Sanchez, R. A.; Patra, T.; Tlahuext-Aca, A.; Strieth-Kalthoff, F.; Glorius, F. DMSO as a switchable alkylating agent in heteroarene C-H functionalization. *Chem. Eur. J.* **2018**, *24*, 10064;

c) Li, J.; Huang, C.-Y.; Han, J.-T.; Li, C.-J. Development of a quinolinium/cobaloxime dual photocatalytic system for oxidative C-C cross-couplings via H₂ release. *ACS Catal.* **2021**, *11*, 14148.

d) Constantin, T.; Górski, B.; Tilby, M. J.; Chelli, S.; Juliá F.; Llaveria, J.; Gillen, K. J.; Zipse, H.; Lakhdar, S.; Leonori, D. Halogen-atom and group transfer reactivity enabled by hydrogen tunneling. *Science* **2022**, *377*, 1323.

Comments (3): In Fig. 5, the authors should add whether 8 and 9 can quench the excited state of the photocatalyst. This has implications for the mechanism that is suggested.

Response: Thanks very much for your suggestions. Fluorescence quenching experiments were conducted for side products **8** and **9**. The results shown that the luminescence emission of photocatalyst couldn't be quenched by **8** and **9** under AcOH/TFE conditions, which agree with our

proposed mechanism (see Supplementary information, Pages S30-S31, supplementary Figures 12-13).

Comments (4): Certain parts of the text should be rephrased with an eye on precision. For instance:

(i) In the introduction, the authors assert with regards to SET and EnT that "in both modes, photocatalysts... directly interact with the substrate, following an "inner sphere-type" mechanism". This is not correct to the best of my knowledge.

(ii) "Notably, this PCES performs as a redox gear for photocatalyst." What does this mean?!

(iii) "Our investigations commenced with evaluating suitable acids that have compatible acidity coefficient" - what does this mean? That the pKa is above or below a certain value?

(iv) "...rules out a long radical chain process". I can infer what the authors are trying to say but more specific phrasing is necessary with regards to the on/off experiments, quantum yield and radical chain processes. As it stands, this generalisation does not make sense.

Response: Thanks very much for your careful examination. We have checked all the manuscript, and corresponding parts of the text were rephrased and revised.

(i) The mentioned sentence "In both modes, photocatalysts directly interact with the substrate, following an "inner sphere-type" mechanism" was deleted (see manuscript, Page 1, the last paragraph).

(ii) In order to better present our work clearly, we revised the term "PCES" as "Relay Electron Transfer". So, the sentence "Notably, this PCES performs as a redox gear for photocatalyst" was deleted. We use other sentences to express the novelty of this approach "Herein, we presented a visible light-induced chemoselective 1,2-diheteroarylation of alkenes from halopyridine, alkene and ring-fused azaarene (Fig. 1c). Notably, the ring-fused azaarenes serve as not only substrate but also an electron-transfer agent for pyridyl radical generation in this protocol." (see manuscript, Page 3, paragraph 2, lines 8-10).

(iii) In the evaluation of acid additives (see manuscript, Figure 2a), we found that weak organic acids, such as BzOH, AcOH, C₂H₅CO₂H and PivOH led to desired product in good yields (85%-91%). And the pK_a values of these four weak acids are about between 4.0 ~ 5.0. But stronger organic acids (pK_a<1.0), such as TsOH and TFA gave bad performance (<30%). The mentioned sentence has been changed to be "Our investigation commenced with screening various Brønsted acid additives for the model reaction of 2-bromopyridine **1a**, TMS-substituted alkene **2a** and 4-

methylquinoline 3a.” (see manuscript, Page 3, the last paragraph, lines 1-2).

(iv) Fluorescence quantum yield experiment was conducted. And the quantum yield of model reaction was determined as 0.87, which might exclude the radical chain process to some degree. The result was added to the manuscript and described as “*Then, light on/off experiments showed that this transformation proceeded only under light irradiation, and the quantum yield of model reaction was determined as 0.87, which might exclude the radical chain process (Fig. 5d).*” (Page 9, Figure 5d and Page 10, paragraph 1, lines 13-15). The detailed procedure for quantum yield calculation was described in Supplementary information (Pages S33-S35).

Comments (5): The authors should make clear what the photocatalysts, Ir-II, Ir-III etc. represent. The corresponding chemical formula should be given.

Response: Thanks very much for your suggestions. The chemical formula of photocatalysts Ir-I, Ir-II, Ir-III and Ir-IV were exhibited in manuscript (Page 4, paragraph 1 and Figure 2b) and Supplementary information (Supplementary Table 1).

Comments (6): Based on the proposed mechanism, it is very hard to understand why styrene does not function in this reaction.

Response: Thanks very much for your comments. The generated pyridyl radical is electrophilic under AcOH/TFE conditions, so the reactivity of styrene is lower than nucleophilic alkyl alkenes. And there are many side reactions (electrophilic substitution) between pyridyl radical with phenyl ring of styrene. Moreover, considering the steric hindrance and reactivity, the benzyl radical from styrene is not feasible for the following Minisci reaction. The explanation was mentioned in Supplementary information (Page S7, paragraph 2).

Response to Reviewer 3:

This reviewer recommended publication of this work in *Nature Communications* after addressing some issues. We gratitude the reviewer's appreciation on our work and we respond to below.

Comments (1): A quantum yield should be provided because light on/off experiments are sometimes not sufficient to determine the reaction mechanism.

Response: Thanks very much for your suggestions. The quantum yield was determined as 0.87, which might exclude the radical chain process to some degree. The result was added to the manuscript (Page 9, Figure 5d; Page 10, paragraph 1, lines 13-15;) and the detailed procedure was described in Supplementary information (Pages S33-S35).

Comments (2): Please comment on the solvent effect and the unsuccessful substrate in supporting information.

Response: Thanks very much for your suggestions. The comments on solvent effect and unsuccessful substrates were added in Supplementary information (Pages S6 and S7).

a) Comments on solvent effect:

As a strongly polar protic solvent, TFE could form a strong hydrogen bond with pyridines and quinolines. The acidity of TFE could improve the electrophilicity of pyridine radical and aza-arene. And TFE could increase the redox potential of ring-fused aza-arenes **3** to be more easily reduced by the photocatalyst (see manuscript, Figure 5b). In addition, TFE has good solubility for all reactants and reagents. Consequently, the model reaction could also provide 20% yield of product in TFE without the use of Brønsted acid (see manuscript, Figure 2e, entry 2).

The above comments were added in Supplementary information (Page S6).

b) Comments on unsuccessful substrates:

Polyhalogenated (2,6 and 3,4-dibromo) pyridines and 3-Br/I pyridines were not suitable for this standard conditions. Amino group was intolerant, and other halogenated aza-arenes like pyrimidine, pyridazine *etc.* couldn't be activated under standard conditions.

The generated pyridyl radical is electrophilic under our conditions, so styrene or alkenes with electron-withdrawing substituents were not suitable for this reaction. α -Functionalized alkenes were also not feasible, properly owing to the instability of the radical intermediate or steric hindrance problem.

Phenyl, ester and halo groups substituted quinolines were not suitable for this reaction. Because electron-withdrawing substituents on *N*-containing aromatic rings might alter the redox potential of the aza-arenes, thereby disrupting the electron transfer. Other aza-arenes, such as pyridine, pyrazine, pyrimidine *etc.* were also reactive, but afforded desired products in low yields.

The above comments were added in Supplementary information (Page S7).

Comments (3): If possible, please isolate and confirm int-B and int-C, as well as the adduct of TEMPO with radical, because HRMS is not enough to confirm the proposed by-products in control experiments.

Response: Thanks very much for your suggestions. The **int-B** was successfully isolated and determined by NMR analysis (see Supplementary information, Page S106). The **int-C** was failed to be isolated, properly owing to its trace yield and too many side-products mixture. The adduct of TEMPO was not detected in its reaction.

Comments (4): Some NMR spectra do not seem to be pure enough or the sampling time is not long enough, such as 4ak, 4na, 4ra, 7i, etc.

Response: Thanks very much for your suggestions. Products **4pa**, **4ra**, **7i**, **7m**, **7o** and **7r** were rescanned using previously stored samples. And products **4ak**, **4fa**, **4ma**, **4na** and **7k** were isolated and purified based on new reactions, so their yields were revised in manuscript and Supplementary

information. All these NMR spectra and data have been updated.

REVIEWER COMMENTS

Reviewer #2 (Remarks to the Author):

I previously reviewed the initial submission of this manuscript to Nature Communications. Since then, the authors have made a number of revisions to address the reviewer comments. Whilst I appreciate the effort that has been made, the proposed mechanism is still deeply flawed. I am not sure the authors have understood the comments which I made last time as the proposal remains thermodynamically infeasible.

This particularly concerns what the authors termed comment 2 from reviewer 2. It is true that other papers also suggest that either DABCO or a protonated heteroarene can act as an electron transfer (J. Am. Chem. Soc. 2015, 137, 8404 & Chem. Eur. J. 2018, 24, 10064). However, all SET steps in these papers are thermodynamically favourable ($E(\text{cell}) = E(\text{red}) - E(\text{ox})$). A number of problems exist in this sense in the authors' manuscript.

1) The photocatalyst is reported as having $E_{\text{ox}} = -0.96 \text{ V vs. SCE}$ according to the second reference above. In their case they measure the protonated quinoline as $E_{\text{red}} = -0.82 \text{ V vs. SCE}$ making the SET favourable. In this case, the authors have measured the same reduction potential (of 3a and AcOH in TFE) as -1.24 V vs SCE . This makes the process unlikely to happen unless the catalyst potential is significantly altered in TFE/AcOH. However this is not discussed at all in the manuscript.

2) This is not the most problematic part of the mechanism. The authors then go on to suggest that reduced 3a (3') can now undergo electron transfer with 1 ($E_{\text{red}} < -2 \text{ V vs SCE}$) where 3' is oxidised back to 3a. How? Where is this energy coming from? E_{cell} is very negative here. Which makes sense if the photocatalyst cannot reduce 1a but it can reduce 3a, how could the reduced form of 3a reduce 1?

Please note this is completely different to the other manuscripts referred to by the authors and in my response. In these cases, the finally reduced (or oxidised) species lies between the potentials of the catalyst and the electron shuttle. Probably this mechanism occurs in this case due to an additional interaction of the electron shuttle with the photocatalyst (e.g. pre-assembly) or because it is present in much greater concentrations.

As this manuscript was originally submitted with this at the centre point of the story, I find it hard to see how this will reach the required standard of Nat. Comms. without serious revision either to address the issue or refocussing the entire point of the manuscript.

Perhaps the authors may like to note that this exact photocatalyst has been reported to undergo ligand modification in situ to form a catalyst able to reach much more reducing states: <https://doi.org/10.1021/jacs.9b07370>

Finally, the phrase "redox gear" is still a nonsense. In part because the authors are trying to explain something that is not feasible (see above). Please remove this phrase when reconsidering the mechanism.

Reviewer #3 (Remarks to the Author):

The authors have addressed all of the issues suggested by this reviewer and the paper is now suitable for publication in this journal.

Response to Reviewer 2:

This reviewer had some concerns and suggestions about proposed mechanism. We gratitude the reviewer's professional comments on our work and we respond to below.

*Comments (1): The photocatalyst is reported as having $E_{ox} = -0.96$ V vs. SCE according to the second reference (Chem. Eur. J. 2018, 24, 10064). In their case they measure the protonated quinoline as $E_{red} = -0.82$ V vs. SCE making the SET favourable. In this case, the authors have measured the same reduction potential (of **3a** and AcOH in TFE) as -1.24 V vs SCE. This makes the process unlikely to happen unless the catalyst potential is significantly altered in TFE/AcOH. However, this is not discussed at all in the manuscript.*

Response: Thanks very much for your comments. The rationality of SET process between protonated quinoline **3** and excited state photocatalyst *Ir-I was explained as the following points:

- (i) According to the results of fluorescence quenching experiments (See manuscript, Figure 2d), the excited photocatalyst *Ir-I was effectively quenched by quinoline [**3a**+AcOH] in solvent TFE instead of [**3a**], [**1a**], or [**1a**+AcOH]. Therefore, AcOH and TFE play important role in this photocatalysis.
- (ii) The excited state *Ir-I was reported as having $*E_{ox} = -0.96$ V vs. SCE, which is a standard value in MeCN (Chem. Mater. **2005**, 17, 5712). However, $*E_{ox}$ is itself dependent on the optical gap $E_{0,0}$ and the ground-state oxidation potential E_{ox} using the formula $*E_{ox} = E_{ox} - E_{0,0}$. In this case, it is possible that there is a large variation in redox properties of photocatalysts in the presence of different solvents, which has been reported that the $*E_{ox}$ of photocatalyst Ir-I could become more negative with increasing solvent polarity (Chem. Sci. **2024**, 15, 3741).
- (iii) To get more accurate results, the $*E_{ox}$ and E_{ox} of photocatalyst Ir-I in TFE were measured according to the reported procedures (Chem. Sci. **2024**, 15, 3741). The ground-state oxidation potential of Ir-I in TFE was determined as $E_{ox} = 0.98$ V vs. SCE (Reported as 1.21 V in MeCN) using the cyclic voltammetry. Besides, UV-vis absorption and steady-state emission spectra of Ir-I in TFE were studied to determine the optical band gap. And $E_{0,0}$ can be calculated using the intersection point between the normalized absorption and emission spectra as $E_{0,0} = 2.52$ eV.
- (iv) Therefore, the redox potential of the excited state *Ir-I in TFE is $*E_{ox} = -1.54$ V ($*E_{ox} = E_{ox} - E_{0,0} = 0.98 - 2.52 = -1.54$ V). The results indicate that the oxidative quenching SET between **3** (for protonated quinoline, $E_{red} = -1.24$ V) and *Ir-I ($*E_{ox} = -1.54$ V) is favourable in TFE ($E_{cell} = E_{red} - *E_{ox} = -1.24$ V + 1.54 V = 0.3 V).

Relevant experimental procedures and calculation methods were added in supplementary information (Pages S42 and S43). And the SET process between *Ir^{III} and protonated quinoline **3** was discussed in manuscript (Page 10, the last paragraph, lines 4-7).

Figure 1. CV spectra obtained for Ir(ppy)₂(dtbbpy)PF₆ in TFE solvent

Figure 2. Optical gap determination for Ir(ppy)₂(dtbbpy)PF₆ in TFE solvent

*Comments (2): The authors then go on to suggest that reduced **3a** (**3'**) can now undergo electron transfer with **1** ($E_{\text{red}} < -2$ V vs. SCE) where **3'** is oxidised back to **3a**. How? Where is this energy coming from? E_{cell} is very negative here. Which makes sense if the photocatalyst cannot reduce **1a** but it can reduce **3a**, how could the reduced form of **3a** reduce **1**?*

Response: Thanks very much for your comments. The proposed mechanism has been revised and illustrated as the following points:

- (i) The CV studies indicate that bromopyridine **1** possesses highly negative reduction potential $E_{\text{red}} < -2 \text{ V vs. SCE}$ under AcOH/TFE standard conditions (See manuscript, Figure 5b). So, it's true that the SET process cannot occur between bromopyridine **1** ($E_{\text{red}} < -2 \text{ V vs. SCE}$) and **3'** or photocatalyst Ir-I. Besides, no additional reducing agent was added in this reaction system.
- (ii) The radical intermediate **3'** is strongly nucleophilic like the α -aminoalkyl radical. And the α -aminoalkyl radical is able to promote the homolytic activation of carbon-halogen bonds via halogen-atom transfer (XAT) (*Science* **2020**, 367, 1021). In this case, intermediate **3'** might serve as halogen abstracting reagent to undergo XAT with bromopyridine **1**, producing the pyridyl radical **Int-I**. This XAT benefits from related kinetic polar effects and manifest the similar reactivity to α -aminoalkyl radical. Probably, an irreversible dissociation of the resulting bromoamine **3''** into the aromatized quinolinium bromide **3** provides the driving force to the process.

Figure 3. Revised possible mechanism and XAT process

- (iii) To verify the XAT reactivity of quinolines **3/3'**, the reactions between quinoline and unactivated alkyl bromides were carried out under the standard conditions. Unactivated alkyl bromides ($E_{\text{red}} < 2 \text{ V vs. SCE}$) are very difficult to go through SET activation under photoredox catalysis. However, as shown in Figure 4, primary and secondary alkyl bromides could react smoothly with quinoline, affording Minisci-type products in decent yields. It is worth noting that there are no common halogen abstracting reagents existing in these conditions, such as organic tin, silicon and amine. And photocatalyst Ir-I is not reductive enough to activate bromoalkenes. Therefore, the results above suggest the possibility of XAT process interacting between quinoline **3'** and

halides, affording carbon radicals.

Figure 4. Minisci reactions of alkyl halides and quinoline

These results were added in supplementary information (Page S43) and discussed in manuscript (Page 11, lines 3-4). The possible XAT interaction between reduced quinoline **3'** and bromopyridine **1** was added in supplementary information (Page S44) and described in manuscript (Page 11, lines 1-2). The proposed mechanism (Figure 6a) and corresponding statements were updated in manuscript.

Comments (3): Finally, the phrase "redox gear" is still a nonsense. In part because the authors are trying to explain something that is not feasible (see above). Please remove this phrase when reconsidering the mechanism.

Response: Thanks very much for your comments. The mentioned phrase in the sentence "Notably, this PCES performs as a redox gear for photocatalyst" has already been deleted in last version of the manuscript.

REVIEWERS' COMMENTS

Reviewer #2 (Remarks to the Author):

The authors have now heavily revised this manuscript when compared to the first version that was submitted with so-called PCES. This has involved major changes in the suggested mechanism which to my view has significantly improved the paper. Although I was critical in earlier rounds of revision, I am pleased that the authors took the comments seriously and think that the most recently proposed mechanism is that which is most consistent with the the experimental data. Whilst there is only circumstantial evidence for XAT occuring from the oxidised, protonated quinoline, this at least provides an explanation that does not contradict thermodynamic data compared to the previously proposed SET mechanism. As the species suggested to undergo XAT is relatively different to those known in the literature, it may have been good to have some more concrete evidence that XAT is possible. However, given that the authors have been careful to word their discussion, I think it is reasonable now for this revision to be accepted at Nature Communications.

Comments:

A small typo is present in Figure 6 "Weekly acidic environment" should be "Weakly acidic environment".

Response to Reviewer 2:

This reviewer recommended publication of this revision in *Nature Communications*.

Comments: A small typo is present in Figure 6 "Weekly acidic environment" should be "Weakly acidic environment".

Response: Thanks very much for your comments. The mentioned typo “Weekly” has been changed to “Weakly” in Figure 6 of manuscript.